# Difficult Vascular Access in Children with Short Bowel Syndrome: What to Do Next?

**DOI:** 10.3390/children9050688

**Published:** 2022-05-09

**Authors:** Chiara Grimaldi, Francesca Gigola, Kejd Bici, Chiara Oreglio, Riccardo Coletta, Antonino Morabito

**Affiliations:** 1Department of Pediatric Surgery, Meyer Children’s Hospital, 50139 Florence, Italy; kejd.bici@meyer.it (K.B.); riccardo.coletta@meyer.it (R.C.); 2Department of Neuroscience, Psychology, Drug Research and Child Health (NEUROFARBA), University of Florence, 50121 Florence, Italy; francesca.gigola@unifi.it (F.G.); chiara.oreglio@unifi.it (C.O.); antonino.morabito@unifi.it (A.M.); 3School of Environment and Life Science, University of Salford, Salford M5 4NT, UK

**Keywords:** children, intestinal failure, central venous catheter, central vascular access

## Abstract

Short Bowel Syndrome and intestinal failure are chronic and severe conditions that may require life-long parenteral nutrition in children. Survival of these children rely on the correct functioning of central venous catheters; therefore, careful management, prevention, and treatment of complications is of paramount importance. Despite a growing awareness of preserving the vascular real estate, a certain number of patients still experience a progressive and life-threatening exhaustion of vascular access. We searched the literature to highlight the current management of children with vascular exhaustion, specifically focusing on vascular access salvage strategies and last-resource alternative routes to central veins. Given the paucity of data, results are reported in the form of a narrative review.

## 1. Introduction

Short Bowel Syndrome (SBS) is a rare and severe condition characterized by the loss of intestinal length leading to intestinal failure (IF) with malabsorption of nutrients and fluids [1]. Causes of SBS may include surgical resection due to inflammatory bowel disease, bowel ischemia/necrosis, trauma, and, in the pediatric population, necrotizing enterocolitis, motility disorders, complications of abdominal wall defects, volvulus, or bowel atresia. There is no consensus in the literature on a precise definition of SBS in terms of intestinal length in children, but intestinal length less than 25% than expected for the gestational age would probably lead to long-term or life-long intravenous supplementation [2]. Moreover, the definition of SBS should not rely on intestinal length alone, because clinical manifestations are mostly dependent on the ability of the remaining intestine to undergo adaptation [3,4].

In SBS patients, the remaining intestine is not able to absorb sufficient amounts of nutrients and fluids to support the patient’s survival and growth [5]. Following an extensive resection, the remaining intestine will undergo a process of adaptation: changes in bowel structure, length, and diameter occur to improve absorption and maximize the contact between the mucosa and nutrients [5]. Subsequently, changes will happen in villus height and crypts’ depth, improving the bowel’s ability to absorb nutrients and fluids [5]. The efficacy of this adaptation process depends on several factors, such as primary diagnosis, residual intestinal length, and function [5,6,7]. Therefore, parenteral nutrition (PN) plays a major role in the management of these patients to maintain an optimal nutritional status and growth rate and to provide them with enough macro- and micronutrients.

In recent years, the use of omega-3-lipid emulsions in PN formulas as well as the institution of Intestinal Rehabilitation Centers has determined great improvement in survival rates in SBS patients [8,9]. Therefore, the need for viable long-term central venous access in affected children has progressively increased, since PN is often required for several years and in some cases for a life-long period.

Given the above, preserving central vascular accesses (CVA) in these patients is essential. PN delivery requires a reliable and sustainable central venous access; in patients who may need PN for short periods of time (2–3 months), peripherally inserted central venous catheters (PICCs) can be used [10]. However, skin-tunneled or subcutaneously implanted central venous catheters (CVCs) in the internal jugular or subclavian vein are recommended for the delivery of long-term PN [11]. Specifically, following the statements of the latest NASPGHAN position paper [12], only tunneled, single lumen, cuffed silicone catheters should be used in children with IF.

When available, an upper extremity access should be the preferred location and the superior vena cava (SVC) should be the first location of CVC insertion, accessed through either the internal jugular veins, the brachial veins, or subclavian veins. If access to these sites is lost, the femoral or saphenous veins can be exploited to reach the inferior vena cava (IVC). Generally, preferred sites of access include the deep veins of the neck and chest (e.g., internal jugular or brachiocephalic veins) or the deep veins of the arm and shoulder (brachial, proximal basilic, and axillary) [13,14]. Other conventional routes, including femoral, subclavian, and cephalic veins, are currently utilized but seem to be associated with higher rates of mechanical and infectious complications [15].

The aim of this review is to collect information regarding the alternative options and management of CVC access exhaustion in patients affected by IF, their outcomes, and their practical use in the clinic.

## 2. Materials and Methods

We performed a literature search of Pubmed to include studies up to February 2022 using the following keywords: central venous access, parenteral nutrition, intestinal failure, unconventional access, hybrid procedure, child, pediatric/paediatric Short Bowel Syndrome.

In addition, reference lists of included articles were screened to identify eligible papers.

Papers were screened by title and abstract by two authors (C.G. and F.G.) and those eligible were read in full text. Disagreement was resolved by consensus. We included studies that described alternative vascular accesses or interventional approaches in pediatric patients affected by IF after loss of all conventional accesses. Only articles published in English in peer-reviewed journals were selected.

We collected information on the management of unconventional vascular accesses, surgical, radiological, or combined approaches and outcomes, when available. Given the heterogeneity of data and limited conclusions drawn by small case series, the results of the search are reported as a narrative review.

### 2.1. Alternative Access Routes for Central Access

#### Inferior Epigastric Vein

The inferior epigastric vein has been described in the past as an alternative access to the IVC in children, even though this vein is frequently reported to be of too small caliber to adapt a CVC. Some sporadic reports on the inferior epigastric vein as an alternative site for central venous access can be found over the decades. Donahoe and Kim were the first to report the use of the inferior epigastric vein in pediatric patients, highlighting two major advantages of this access: the proximal end of the catheter can be tunneled up above the diaper, preventing contamination, and the baby can assume its natural supine position with the legs flexed without kinking the catheter [16].

More recently, Singh and Martin and Saleh et al. [17,18] reported their experience and described some useful tricks to optimize this access and prevent the migration of the tip of the catheter inferiorly in the groin or into one of the major tributaries [17]. Among a total of 54 patients, including adults and children, only 7 catheters were removed because of an infectious complication, while no cases of thrombosis were reported. Average longevity of the catheters was 18.5 months [17,18].

### 2.2. Gonadal Vein

Pérez Illidge et al. [19] recently reported a single-center series on the use of the gonadal vein as a non-conventional, last-resource vascular access. Similar reports on the use of gonadal veins to reach the IVC in patients who previously exhausted all other options can be found in the literature over the last decades [18,20,21]. The major issue is the need for a direct access to the peritoneal cavity in patients with previous multiple abdominal surgeries and who would likely need other laparotomies (specifically for intestinal transplantation). To overcome this limit, Saleh et al. [18] described a retroperitoneal approach to place a long-term catheter in the right gonadal vein in children who experienced extensive thrombosis of the iliofemoral system and lower IVC. They performed the procedure in five children and reported no complications during a medium-term follow-up (10 months). Some authors have reported on the positioning of a gonadal access during the intestinal transplant surgery, considering it a very unstable access and thus comparable to a short-term access, useful to overcome the immediate post-surgery period [19].

### 2.3. Intercostal and Other Thoracic Veins

When considering thoracic veins, multiple insertion routes may be used. Some catheters could be inserted throughout a percutaneous approach, others via more aggressive techniques, such as thoracoscopy or thoracotomy.

Access to the heart may be obtained using the azygos vein reached via the intercostal veins: this approach was first defined by Newman et al. [22] in the 1980s. In 2005, Tannuri et al. [23] described 2 cases in which a direct insertion of a Port-a-Cath via the intercostal veins through a thoracotomy approach was reported with no complications. The two catheters lasted for 6 and 13 months, respectively. They also illustrated the cannulation of the third intercostal vein in a 16-month-old patient with thrombosis of the superior and inferior vena cava. Likewise, Saleh et al. in 2008 described the use of intercostal veins as direct access to the right atrium in five pediatric patients using a thoracotomy approach. The advantage of this technique is that multiple intercostal veins can be cannulated allowing for different CVAs.

A thoracoscopy-assisted long-term CVA positioning via the direct punction of the azygos vein was first reported in a child in 2008 by Sola et al. [24]. A modification of this approach was more recently described in one young adult who, while on the waiting list for intestinal transplantation, took advantage of a double long-term access, placed with a combined percutaneous and thoracoscopic approach, via puncture of 2 different intercostal veins and the insertion of the central lines into the heart through the azygos vein [25]. Direct placement of a catheter in the azygos vein has been performed successfully by the same surgical team. Unfortunately, one of the limitations of this approach is that guide-wire substitution of the catheter in case of malfunctioning may be challenging or impossible [24].

The use of the internal mammary vein for central venous access was detailed by Alomari et al. [26,27]. They reported their experience with percutaneously inserted catheters in 8 children: 6 children out of 8 were affected by IF. All procedures were performed under general anesthesia in an interventional radiology environment, using a sterile Seldinger technique. Catheter placement was successful in 5 IF cases via the internal mammary vein, while in 1 case the procedure failed due to the occlusion of the superior vena cava. Five patients underwent catheter exchange procedures due to malfunction, malposition, and infection. The mean reported dwell time/patient is 314 days. The authors argue that this vein is usually of small caliber in children and that it should be considered as an alternative route only when dilated because of central venous obstruction.

### 2.4. Transhepatic Access

In case of exhaustion of access sites and superior and inferior vena cava thrombosis, hepatic vein cannulation could be a viable option. Unfortunately, transhepatic access is associated with high mechanical instability of the CVC, high short-term and long-term complication rates, and risk of infection [28,29,30]. Possible dislodgment is induced by the patient’s movements and breathing and it is a major concern because bleeding into the peritoneal cavity may follow [31,32]. Given an expected shorter life span, the use of these devices is usually limited to very selected patients, usually children awaiting for intestinal transplant [33,34].

The use of the transhepatic approach to central veins in pediatric patients was first described in the early 1990s by two different teams [35,36]. Out of 12 children, the authors reported 3 cases of catheter dislodgment, 2 cases of catheter exchange, and 6 episodes of CLABSI [35].

De Csepel et al. [37] reported two cases of transhepatic cannulation in pediatric patients using ultrasound to identify the right or middle hepatic veins and a percutaneously inserted guidewire to advance the CVC into the right atrium. In one child the device had no sign of dislocation at long-term follow-up (9 months), while the other patient suffered from an accidental dislocation and several episodes of infection that led to the removal of the CVC. Patient selection is extremely important: one case of long-lasting and last-resource transhepatic access is reported by Diamanti et al. [38] in a boy with IF and an absolute contraindication to intestinal transplantation. Following the original technique described by Sharif et al. [32], who used an open surgical access to the liver in order to fix the CVC directly to the capsule, a hybrid surgical and radiological procedure was performed to maximize the stability of the central line with direct fixation of the catheter to the liver capsule and to the abdominal wall. Mortell et al. [29] reported 5 cases of transhepatic cannulation of the right atrium: of these, one patient needed long-term parenteral nutrition for necrotizing enterocolitis, while the others suffered from congenital cardiac diseases. The mean catheter life was 98.8 days with a mean follow-up of 9 months (range, 5–18 months). Two devices were removed because of CVC infection and the erosion of the wall of the right atrium by the catheter tip.

Moreover, the transhepatic access should not be considered a suitable approach in patients who require a combined liver–intestine transplantation due to the need to remove the catheter during transplantation [33].

### 2.5. Direct Access to the Heart

Even if very rarely used, direct intra-atrial catheters have proved adequate during uncomplicated pediatric intestinal transplantation [33,39]. This route has been described in adults associated with multiple complications, such as pleural effusion secondary to CVC displacement [40] or CVC migration [41]. Thoracoscopy can provide direct vision of the heart, improving the long-term stability of the device [42].

Detering et al. [43] described a successful direct atrial insertion of a central line in a 11-year-old girl via a thoracotomic approach with no complications at a 3-year follow-up. Rodrigues et al. [33] reported 4 cases of direct intra-atrial catheter positioning in pediatric patients referred for small bowel transplantation. In three cases the catheter underwent dislodgement and in one case the patient suffered from left pleural effusion, requiring a chest drain.

Given the high risk of CVC dislocation and the high burden of life-threatening complications, this access should be considered only in the case of exhaustion of all other options.

### 2.6. Some Very Unusual Accesses

Translumbar approach to the IVC has been reported in adults, with a high rate of dislocation and IVC thrombosis, sometimes associated with a caval filter [44]. In 1992, Azizkhan et al. [35] described the use of percutaneous translumbar IVC catheters for prolonged venous access in 4 children, all PN dependent. In one case, the catheter was dislodged following the child’s growth, in one case it was replaced due to catheter thrombosis, and one patient had multiple episodes of central line associated bloodstream infections (CLABSI), successfully treated with antibiotics. Malmgren et al. [45] described the translumbar access in 4 children with no procedure-related complications; catheters were still in place 4.8 months after placement.

Historically, the use of surgical arteriovenous fistulas like those used for hemodialysis has been reported in adult patients to deliver total parenteral nutrition [46]. More recently, the approach gained some popularity, and its use has been proposed in adults needing long-term TPN [47]. The authors advocate a good patency rate and the need to explore this technique in patients with expiration of vascular real estate despite some procedure-specific complications such as cardiac overload. Unfortunately, due to the small caliber of vessels in children, no reports on this technique are available in the pediatric population.

### 2.7. Salvage of Central Venous Lines

A variety of methods to salvage CVC and recanalize obstructed vessels are reported in the literature [48,49,50,51]. For patients with obstructive thrombosis in whom systemic thrombolysis fails or is unsuitable, aspiration thrombectomy or catheter-directed thrombolysis may be effective options. [52]. Multiple options are currently available to recanalize thrombosed vessels, starting from less invasive techniques (such as guidewires and dilators, angioplasty balloons) stepping up to aggressive options such as sharp recanalization, cutting balloons, or permanent stenting [53,54,55]. Advanced treatment options usually combine pharmacological and mechanical thrombolysis. Both options, when successful, avoid catheter replacement. However, the use of thrombolytic agents, either systemically or locally, can be unsuccessful, especially in the case of long-standing catheters [56,57]. Moreover, pharmacologic treatment potentially increases the risk of hemorrhage. Combined radiological and pharmacological procedures seem to improve the success rate. For example, combined use of thrombolytic drugs and interventional radiology techniques such as balloon angioplasty, stent placement, and endovascular repositioning of a catheter, have proven to be successful and reduce the catheter changing rate [58,59,60]. Treatment options to recanalize obstructed vessels are detailed in Table 1.

Recently, reports of hybrid approaches for catheter salvage in children have increased, and data are less anecdotal. In 2018, Sieverding et al. [61] reported the results of their systematic and comprehensive strategy to approach patients with challenging vascular access. A careful anatomical assessment, associated with the sensible use of all the surgical and interventional radiology armamentarium, allowed the authors to ensure a permanent vascular access in a large group of otherwise extremely challenging patients. Similarly, de Buys Roessingh et al. [62] reported on 2 children, 10 and 11 months of age, respectively, who underwent a successful combined endovascular and surgical recanalization after central venous obstruction. In both cases, patients had developed thrombosis of the SVC and right and left brachiocephalic veins; previous local thrombolytic therapy using recombinant tissue plasminogen activator was unsuccessful. Despite the presence of the thrombus, a vascular sheath and the guide wire were inserted in the right internal jugular vein and the right medial thyroid vein under direct radiological control. A central line was then placed on the guide wire. In both cases, no complications occurred during the procedure. The first child underwent two further similar procedures for replacement of the central line in the following three years due to thrombosis. In the second case, the catheter was still in place with no complication after a one-year follow-up.

Mehta et al. [63] described a stereotactic technique of catheter placement in 5 children, successfully performed by the interventional radiologists. The authors reported two complications: an accidental breach of the pericardium and one unexplained death 24 h after the procedure, respectively.

## 3. Discussion

Management of vascular access needs to be part of a comprehensive and multidisciplinary strategy aiming, as a first step, to preserve the vascular real estate of children requiring long-term PN. Therefore, a wise and careful planning of vascular accesses associated with a detailed knowledge of the patient-specific vascular anatomy since the beginning of the child’s clinical history is mandatory.

Actually, in highly specialized multidisciplinary teams, up to 90% of catheters are placed by interventional radiologists [19]. All children with IF should benefit from the expertise of a hybrid, multidisciplinary, combined surgical and radiological team from the beginning of their clinical history. Furthermore, this approach should be mandatory for those children experiencing a progressive exhaustion of their CVAs despite all attempts.

Progressive loss of vascular access is the consequence of multiple, intertwined factors, mostly central line infections and vascular thrombosis.

CLABSI are a major cause of morbidity and mortality in PN-dependent patients [64]. The literature reports a highly variable incidence in CLABSI rates [10]: a systematic review by Dreesen et al. in 2013 reported a range of 0.38–4.58 per 1000 catheter days among the adult population, while Chu et al. reported an incidence of 8.6 per 1000 catheter days in children [64,65]. Prevention of CLABSI starts with family education, training, use of prevention bundles on CVC insertion sites, and catheter-care maintenance protocols [66]. In SBS patients, multiple studies demonstrate the efficacy of a taurolidine-citrate-heparin catheter lock in determining a clinically substantial and cost-beneficial reduction in CLABSI occurrence [67,68,69].

Even in correctly managed patients, life-long need for PN may lead to progressive loss of conventional access routes, such as the axillary, external jugular, internal jugular, subclavian, saphenous, brachio-cephalic, and femoral veins. These veins can become unavailable due to stenosis and/or thrombosis. Regardless of the underlying pathology requiring long-term central lines, the overall reported rate of thrombosis and stenosis in children varies from 26% to 75% [70,71,72]. Thrombotic occlusion of the superior and inferior vena cava may occasionally occur, thus compelling to find direct and challenging access to the heart.

Prevention and/or aggressive treatment of these complications are essential to ensure a long-term functioning of chronic vascular access in patients who totally depend on these devices. For this reason, over the decades, specific guidelines, such as the comprehensive ESPEN guidelines, have been produced to maximize the longevity of vascular access and to prevent the exhaustion of the vascular real estate [11].

Unfortunately, despite a growing awareness of the importance of preservation of vascular assets, loss of standard access sites is a common clinical challenge, especially in children with chronic IF, who are dependent on PN. Loss of one central vein can be documented in the majority of children with intestinal failure (57%), while loss of two or more central veins is reported in 40–46% [73,74]. Critical loss of vascular accesses with progressive exhaustion of the vascular real estate, is, per se, one of the indications to intestinal transplantation, according to the American Gastroenterology Association (AGA) [75] and ESPEN guidelines, respectively [76]. The latest ESPEN indications to intestinal transplantation are summarized in Table 2.

First-line management of long-term catheter needs to focus on punctual protocols for prevention and treatment of complications: infection, dislocation, device deterioration, and obstruction.

Efforts should aim to salvage the actual catheter or, when the device needs to be changed, to reuse the same vessel by guide-wire substitution of the central line or by more aggressive techniques such as pharmacological (systemic and/or local) or mechanical thrombolysis, if required. Over the decades, multiple techniques for vessel reutilization emerged, mostly from the experience of adult endovascular radiologists. Therefore, children should benefit from these expertise and data should be collected to elaborate systematic guidelines.

To overcome life-threatening complications, alternative routes have been developed and further implemented, as detailed in Table 3. In the last decades, some helpful solutions have been learned from the vast armamentarium broadly used in the adult population.

Moreover, discussion by experienced and multidisciplinary teams with specific competences is mandatory when it comes to the use of alternative access sites. The decision to proceed to use alternative routes should come after implementation of strategies aiming to treat the venous obstruction or stenosis by all possible means (balloon dilatation, mechanical thrombolysis, stenting).

Alternative sites or salvage procedures should be proposed only by experienced teams in accordance with an intestinal transplant center.

Obviously, despite these attempts, alternative routes for central line access will always be necessary for a small number of children. Technical improvements allow safe access of unusual veins; therefore, surgeons should be aware of these techniques to deal with rare but very challenging anatomical conditions. When extensive thrombosis of the larger vessels prevents the placement of a new central line, intercostal, hepatic, gonadal, and inferior epigastric veins should be kept in mind as last-resource but safe and effective routes.

## Figures and Tables

**Table 1 children-09-00688-t001:** Options of treatment for recanalization of thrombosed vessels.

Treatment Options
Thrombolytic agents
Guidewires substitution
Angioplasty balloon dilatation/cutting balloon
Sharp recanalization
Endovascular stent placement

**Table 2 children-09-00688-t002:** Indications for intestinal transplantation adapted from ESPEN guidelines [76].

Evidence of Advanced or Progressive Intestinal Failure—Associated Liver Disease
Hyperbilirubinemia > 75 μmol/L (4.5 mg/dL) despite intravenous lipid modification for >2 months
Elevated serum bilirubin and/or reduced synthetic function (subnormal albumin or elevated international normalized ratio), and/or laboratory evidence of portal hypertension and hypersplenism persisting for >1 month in the absence of confounding events
In children hrombosis of 3 discrete upper body central veins (left subclavian and internal jugular, right subclavian and internal jugular) or occlusion of a brachiocephalic vein (in adults evaluate on a case-by-case basis)
Life-threatening morbidity in the setting of indefinite PN dependence, as suggested by: In children, 2 admissions to an intensive care unit after diagnosis of intestinal failure because of cardiorespiratory failure (mechanical ventilation or inotrope infusion) due to sepsis or other complications of intestinal failure; In adults, on a case-by-case basis
Invasive intra-abdominal desmoids in adolescents and adults
Acute diffuse intestinal infarction with hepatic failure
Failure of first intestinal transplant

**Table 3 children-09-00688-t003:** Types of central vascular access [16].

Conventional Accesses	Non-Conventional Accesses	Last-Resource Accesses
Jugular	Azygous	Transhepatic
Subclavian	Translumbar	Direct right atrial insertion
Femoral	Intercostal veins	Gonadal vein
	Mammary	
	Arteriovenous fistula	

## Data Availability

Not applicable.

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
