# Peer review of "Difficult Vascular Access in Children with Short Bowel Syndrome: What to Do Next?"

_children, 2022, doi:10.3390/children9050688_

Round 1

Reviewer 1 Report

Thank you for this thorough review of this complex and important issue, a good compilation and guide to clinicians dealing with short bowel syndrome.

Author Response

thank you for your kind comments

Reviewer 2 Report

  • In the Introduction, Page 1; Para 2: would add ‘Motility disorders’ to the causes of Intestinal Failure (IF)
  • Page 1; para 3 ,line 39: would add: bowel structure, length and diameter…
  • Page 5; para 2; line 180: …via punction of 2 different…; should perhaps be ‘via puncture of 2 different’…

Author Response

  • In the Introduction, Page 1; Para 2: would add ‘Motility disorders’ to the causes of Intestinal Failure (IF). thank you for your comment. paragraph amended accordingly
  • Page 1; para 3 ,line 39: would add: bowel structure, length and diameter… Thank you for your comment. paragraph amended accordingly
  • Page 5; para 2; line 180: …via punction of 2 different…; should perhaps be ‘via puncture of 2 different’… thank you for your comment. paragraph amended accordingly

Reviewer 3 Report

In the introduction part of the manuscript on line 43, the abbreviation ''PN'' should be explained. I think "parenteral nutrition".

The introduction part of the manuscript is written quite long. I think it would be more appropriate to shorten this section and add most of what has been written here as a "discussion'' section at the end of the manuscript.

King regards.

Author Response

In the introduction part of the manuscript on line 43, the abbreviation ''PN'' should be explained. I think "parenteral nutrition". thank you for your comment. amended in paragraph

The introduction part of the manuscript is written quite long. I think it would be more appropriate to shorten this section and add most of what has been written here as a "discussion'' section at the end of the manuscript. 

Dear rewever, thank your for the suggestion: we changed the original manuscript accordign to your comments